# A Pig Mass Estimation Model Based on Deep Learning without Constraint

**DOI:** 10.3390/ani13081376

**Published:** 2023-04-17

**Authors:** Junbin Liu, Deqin Xiao, Youfu Liu, Yigui Huang

**Affiliations:** College of Mathematics Informatics, South China Agricultural University, Guangzhou 510642, China

**Keywords:** computer vision, deep learning, mass measurement, convolutional neural network

## Abstract

**Simple Summary:**

Constructing a contactless pig mass estimation method through computer vision technology can help us to adjust our pig breeding program to improve production efficiency. Due to the complexity of the actual production environment, there are few reports on pig mass estimation in an unconstrained environment. In this study, we constructed a pig mass estimate model based on deep learning without constraint. The experimental results proved that the pig body mass estimation model constructed in this paper can quickly and accurately obtain the body mass of pigs. The model can evaluate the body quality of sows in real-time in an unconstrained environment, thereby providing data support for grading and adjusting breeding plans, and has broad application prospects.

**Abstract:**

The body mass of pigs is an essential indicator of their growth and health. Lately, contactless pig body mass estimation methods based on computer vision technology have gained attention thanks to their potential to improve animal welfare and ensure breeders’ safety. Nonetheless, current methods require pigs to be restrained in a confinement pen, and no study has been conducted in an unconstrained environment. In this study, we develop a pig mass estimation model based on deep learning, capable of estimating body mass without constraints. Our model comprises a Mask R-CNN-based pig instance segmentation algorithm, a Keypoint R-CNN-based pig keypoint detection algorithm and an improved ResNet-based pig mass estimation algorithm that includes multi-branch convolution, depthwise convolution, and an inverted bottleneck to improve accuracy. We constructed a dataset for this study using images and body mass data from 117 pigs. Our model achieved an RMSE of 3.52 kg on the test set, which is lower than that of the pig body mass estimation algorithm with ResNet and ConvNeXt as the backbone network, and the average estimation speed was 0.339 s·frame^−1^ Our model can evaluate the body quality of pigs in real-time to provide data support for grading and adjusting breeding plans, and has broad application prospects.

## 1. Introduction

Pig body mass is an important indicator for measuring the growth and development of pigs and assessing their health status [1,2]; it plays an important role in pig breeding. Variations in animal body mass determine the production management strategy of the animal, as pigs being too fat or too lean can affect their selling price. To improve feeding efficiency, commercial farms need to develop appropriate feeding programs based on the body mass growth curve of pigs [3,4,5]. Therefore, an effective method for intelligent determination of pig body mass can help feed managers to make more rational feeding decisions [6,7,8,9].

Current pig farms rely on farm staff to haul pigs to weighing scales for weighing. This method is not only labor-intensive, it can have harmful effects, such as causing animal stress, injury to staff and animals, and even increased pig mortality [10,11]. Although direct weighing provides the most accurate pig body mass, the process is tedious and very time-consuming, usually taking 3–5 min for farm staff to weigh a pig; frequent interaction between the scale and the pig may even cause damage to the machine, resulting in incorrect results [12,13].

Non-contact pig body mass determination using computer vision technology can improve the health status and welfare of animals, and has become a hot spot for research applications in recent years [14,15]. Computer vision-based pig body mass assessment is a non-invasive, fast, and convenient method that can reduce stress on pigs and farm staff, help farms to continuously monitor pig body mass in real-time, and adjust farming strategies promptly. Pezzuolo et al. [16] used a 3D camera to obtain mass-related parameters such as pig chest circumference, body length, and body height, and established a nonlinear pig body mass model with a determination coefficient R^2^ over 0.95. Jun et al. [17] extracted features such as body area, head position, and body twist degree of pigs in a top-view 2D picture by feature engineering, and determined final pig body mass using a fully connected neural network with a determination coefficient R^2^ of 0.79. Shahinfar et al. [18] compared four machine learning algorithms, namely, multilayer perceptron, model tree, random forest, and support vector machine, in combination with sequential minimum optimization to construct a beef cattle body mass estimation model based on a physiological characterization information series obtained from beef cattle, and found that the model using support vector machine combined with sequential minimum optimization worked best. Le et al. [19] used a 3D camera to extract morphological features such as the volume and surface area of cattle and constructed a regression equation to determine the body mass of cattle, with a coefficient of determination R^2^ of 0.93. Though he above studies based on machine learning methods have achieved results in animal body mass estimation, machine learning-based algorithms need to be constructed for different datasets, and the model construction process is tedious.

Compared with the traditional machine learning methods that require the construction of feature engineering, the automatic learning of extracted features using deep learning has achieved better results in building animal body mass determination models [20,21,22,23]. Convolutional Neural Networks (CNN) are a class of Feed-forward Neural Networks (FNN) that include convolutional computation. CNNs have a deep structure; they are one of the representative algorithms in deep learning, and are often used to analyze visual images [24]. The model constructed by Dan et al. [25] based on convolutional neural networks made great progress in determining the body mass of a single pig, with an average percentage error of 7% and a coefficient of determination R^2^ of 0.95. The method of Cang et al. [26] added a regression branch to Faster R-CNN to integrate the pig target detection and body mass regression networks into the end-to-end network, thereby improving the generalization performance of the model. Zhang et al. [27] controlled pigs in a confinement pen, acquired pig image data through a depth camera, and constructed a multi-output regression convolutional neural network to estimate pig body mass using a modified Xception network for feature extraction; using this approach, the R^2^ of the coefficient of determination of the weight estimation model was increased to 0.988. Meckbach et al. [28] constructed a pig body mass estimation model based on EfficientNet and used a preprocessing algorithm to process the images before data input. This process was able to reduce the error caused by shaking of pigs in the confinement pen, and the average percentage error on the test set was 4%. He et al. [29] borrowed the MHSA structure of BotNet to improve ResNet and constructed a body mass estimation model on this basis. Most of the above studies were conducted by image acquisition and body mass measurement of pigs in confinement pens with limited data sample scenarios and more restrictions on the pigs. In a slaughterhouse, for each batch of pigs the staff need to determine the grade of the pigs according to their body mass in order to separate the pigs into pens; thus, an unconstrained body mass estimation model that can be applied in the aisle can help slaughterhouse staff to complete the grading of pigs more efficiently. In the aisle environment, pig movement and light changes affect the quality of the captured images; when pigs pass through more densely, it is more difficult to extract individual pigs, as the presence of mutual obstruction and fences may cause pigs’ bodies to be incomplete in the images, affecting the accuracy of the body mass estimation model. The currently available studies have neither discussed nor solved the problems of unstable image quality, difficulty in extracting individuals, and occlusion that inevitably occur in unconstrained environments. Therefore, the current solutions for pig body mass estimation in conventional unconstrained environments remain very flawed.

To realize pig body mass determination in an aisle environment and reduce the effect of occlusion on pig body mass determination, this paper combines a data pre-processing algorithm, instance segmentation, and keypoint detection algorithm, and proposes a Pig Mass Estimation Model based on Deep Learning Without Constraint (PMEM-DLWC). The proposed model improves data quality and reduces the impact of abnormal data on model training through data pre-processing algorithms such as blurry detection and similarity metrics; moreover, it extracts complete individual pigs while dealing with occlusion problems through instance segmentation and keypoint detection. Our proposed approach realizes the estimation of pig mass using low-cost monitoring equipment, providing a solution for unconstrained pig mass estimation.

## 2. Materials and Methods

The technical roadmap in the specific experiment of the research content of this paper is shown in Figure 1. First, we constructed a database using video image data captured by a TB-1217A-3/PA camera and pig mass data obtained from weighing in the aisle scene of the slaughterhouse. We then used blur detection and measured the images’ structural similarity index to improve the dataset quality. Second, using the RGB images and pig weight data, we developed a pig mass estimation model by combining the Mask R-CNN instance segmentation algorithm, the Keypoint R-CNN keypoint detection algorithm, and the ResNet-based body mass determination algorithm. We improved the backbone network structure, activation function, and normalization method of each algorithm in the model to achieve accurate pig weight estimation.

### 2.1. Data Collection Scenarios

The data used in this paper were collected from June to August 2021 at the slaughterhouse of Wen’s Group in Heyuan City, Guangdong Province. The experimental scenario is shown in Figure 2, where a portion of pigs from each batch arriving at the slaughterhouse was randomly selected and marked on their backs, then weighed using a weighing scale with a weighing accuracy of 0.1 kg. We collected 117 pigs’ video image data and body mass data. Considering the errors caused by the pigs’ activities, each pig was weighed five times and the average value was taken as the real body mass of each pig. The body masses of the pigs were distributed between 70–160 kg, and all pigs belonged to the same breed. The farm staff cleaned and calibrated the weighing scales every day to ensure that the body masses of the pigs were obtained correctly. After weighing, we pulled the pigs into the aisle, which is about 1.5 m wide, for free movement. The camera was installed above the aisle to collect video data when the pigs were walking in the aisle; the camera was parallel to the ground and 2.8 m away from the ground. The camera was a Hikvision TB-1217A-3/PA binocular camera (Hikvision, Hangzhou, China) capable of acquiring color and thermal infrared data and with a video frame rate of 25 fps and a resolution of 2688 pixels × 1520 pixels. The camera was connected to a microcomputer (Dell OptiPlex 3080 Micro, Windows 11 Pro, Intel Core i5-10500T, 8GB DDR4 RAM, 256GB SSD) to ensure timely data upload to the cloud.

### 2.2. Improving Data Quality

In this paper, the video frames from the captured video data were used as the training dataset for the body mass determination model. Due to the movement of the pigs and changes in lighting, the acquired images may contain blur, which can affect the training process of the body mass determination model. To select images with high quality for model training, this paper used a Laplacian operator-based image blur detection algorithm to evaluate the clarity of the images and improve the quality of the dataset while at the same time ensuring sufficient data for training. To avoid too many similar video frames in the dataset, a similarity detection algorithm based on structure similarity index measure (SSIM) was used to eliminate the images with too much similarity.

#### 2.2.1. Image Blur Detection

The quality of images has a great impact on the accuracy of a body mass estimation model. In our experiments, we found that pigs passing through the aisle at a uniform speed (v < 2 m/s) would not have an effect on the image quality, while the images captured by the camera were blurred when the pigs were frightened and passed through quickly. To ensure the quality of the images, in this paper we used a blur detection algorithm based on the Laplacian operator to evaluate the sharpness of the images. The Laplacian operator is used to convolve the image to obtain the high-frequency components of the image, then the high-frequency components of the image are summed and the sum of the high-frequency components is used as the image sharpness evaluation criterion. The definition of the Laplacian operator and the definition of image sharpness based on the Laplacian gradient function are shown below.
(1)$$L(x,y)=\frac{\partial^{2}f(x,y)}{\partial x^{2}}+\frac{\partial^{2}f(x,y)}{\partial y^{2}}$$
(2)$$D(f)=\sum_{y}\sum_{x}\left|L\left(x,y\right)\right|$$
where *f*(*x*,*y*) is the pixel value of pixel point (*x*,*y*) on the image and *D*(*f*) is the Laplacian function value of the image. The Laplacian operator is often used for edge detection to highlight regions of an image that contain rapid gradient changes’ a smaller value of the Laplacian function indicates that there are fewer edges in the image, meaning that the image is more blurred. The distribution of Laplacian function values for the original dataset is shown in Figure 3. Considering the range of Laplacian function values associated with the dataset, the sharpness values were scaled to between [0, 1] using the maximum–minimum normalization method to define a common threshold value for all images. To ensure that the images for body mass estimation were clear and there are enough images for body mass estimation, in this paper we determined the sharpness threshold using the following equation to filter images based on sharpness.
(3)$$Q(\beta )=\max(\log(N(\beta )\cdot G(\beta )))\beta \in \left(0,1\right)$$
where *β* is the normalized image sharpness value, N(*β*) indicates the number of images with a sharpness value greater than *β*, and G(*β*) indicates the average value of the sharpness of images with sharpness greater than *β*. After calculation, the value of Q(*β*) reaches the maximum when *β* = 0.64; thus, we chose *β* = 0.64 as the threshold value for determining whether an image was clear or not.

#### 2.2.2. Structural Similarity Index Measure

In the studies of Meckbach et al. [28] and Hansen et al. [30], the similarity between images was measured using the structural similarity index measure (SSIM) to avoid the presence of too many similar images in the dataset. In this paper, we followed this idea for discrimination of the video frames obtained from the same video interception using the SSIM metric. The formula of the SSIM metric is as follows:(4)$$SSIM(x,y)=\frac{\left(2\mu_{x}\mu_{y}+c_{1})(2\sigma_{xy}+c_{2}\right)}{\left(\mu_{x}^{2}+\mu_{y}^{2}+c_{1})(\sigma_{x}^{2}+\sigma_{y}^{2}+c_{2}\right)}$$
where *x* and *y* represent two images, with *μ_x_* being the pixel mean of *x*, *μ_y_* the pixel mean of *y*, *σ_x_^2^* the variance of *x*, *σ_y_^2^* the variance of *y*, *σ_xy_* the covariance of *x* and *y*, c_1_ = (0.03 *L*)^2^ and c_2_ = (0.01 *L*)^2^ being constants used to maintain stability, and *L* being the dynamic range of pixel values.

For video frames belonging to the same video, one image was randomly selected as the initial dataset, after which the images added to the dataset were subjected to SSIM calculation with all the images in the dataset; only those images that did not exceed the threshold were added to the dataset. Figure 4 shows the effect of different SSIM thresholds on the number of selected video frames; as shown in the figure, when the SSIM threshold is less than 0.8, the number of video frames obtained from most video screenings is less than 5. To select enough video frames in each video and avoid too much similarity between video frames, in this paper we choses SSIM = 0.85 as the threshold value.

### 2.3. Construction of Dataset

After quality enhancement of the original data, the instance segmentation dataset, keypoint detection dataset, and body mass dataset for training different algorithms were constructed based on the above pig video image data and body mass data. We used the Labelme annotation tool for the needs of deep learning modeling in this paper. The images in the dataset consisted of video frames. All datasets were randomly divided into a training set, test set, and validation set using a ratio of 7:2:1, and it was ensured that the pigs in the training set would not appear in the validation set or test set.

### 2.4. Construction of Pig Mass Estimation Model Based on Deep Learning without Constraint

The unconstrained pig body mass estimation model in this paper consists of three algorithmic sub-modules, including a Mask R-CNN-based pig instance segmentation algorithm, Keypoint R-CNN-based pig keypoint detection algorithm, and improved ResNet-based pig body mass estimation algorithm. The process of the model is shown in Figure 5: for the input pig images, the pig instance segmentation algorithm is first used to extract the masks of different individual pigs in the images and remove irrelevant objects according to the masks; next, keypoint detection is performed on the pigs by the pig keypoint detection algorithm, eliminating obscured pig images according to the integrity of the keypoints; finally, these compliant images are binarized and the binarization results are input into the pig body mass estimation algorithm for feature extraction and calculation. To improve the robustness of the model, the constructed model was enhanced using the transfer learning method, with the instance segmentation algorithm using a model trained on the COCO dataset as the initial model. The weight model of the pig instance segmentation algorithm was used as the pre-trained model for the keypoint detection algorithm, and the weight model of the keypoint detection algorithm was used as the pre-trained weight model for the body mass estimation algorithm.

#### 2.4.1. The Pig Instance Segmentation Algorithm Based on Mask R-CNN Network

Unlike the restriction pen scenario, where pig movement is restricted and data are collected for only one pig, in the aisle scenario there are usually multiple pigs passing through at the same time and the phenomenon of pigs sticking together occurs frequently. In the pig body mass estimation algorithm, the network needs to learn the feature information related to the body mass of the target individual pig. To obtain the complete target information of an individual pig, this paper obtains the mask of the pig in the image through the instance segmentation algorithm to extract the individual pig from the image. The instance segmentation effect is shown in Figure 6.

The pig instance segmentation algorithm in this paper was constructed based on Mask R-CNN [31]. Mask R-CNN consists of two parts: Faster R-CNN for object detection and classification, and a fully connected network for semantic segmentation. The advantage of Mask R-CNN over traditional segmentation methods is that it can perform detection, classification, and image segmentation simultaneously; the structure is shown in Figure 7.

While Mask R-CNN has good segmentation for relatively significant large objects, the similarity between pigs is high and the contours and boundaries of densely passed pigs are less distinct, which interferes with segmentation of the pigs. Therefore, the instance segmentation algorithm needs to classify the pixels in the image while detecting individual pigs, which requires the algorithm to have a strong ability to handle multi-scale information. In this paper, we combined the backbone ResNet network [32] of Mask R-CNN with recent advances in convolutional neural networks to improve the network’s ability to handle multi-scale information. The main improvements include the following five aspects.

(1)Multi-branch Convolution Module. To address the shortcomings of ResNet in terms of its poor ability to represent fine-grained features, our model borrows the idea of multi-branch convolution from Res2Net [33] and uses one group of smaller filter sets to replace the 3 × 3 filters, as shown in Figure 8. The new 3 × 3 filter sets are connected in a hierarchical residual-like style to increase the perceptual field of the network and improve its multi-scale representation capability. This is accomplished as follows. After 1 × 1 convolution, the input feature maps are divided into multiple groups; one group of filters first extracts information from one set of input feature maps, then the output feature maps of the previous group are sent to the next group of filters along with another set of input feature maps. This process is repeated several times until all input feature maps are processed. Finally, the feature maps of all groups are connected and sent to the 1 × 1 filter to completely fuse the information. Along any possible path from the input feature map to the output feature map, the equivalent perceptual field is increased when passing through the 3 × 3 filter, resulting in more equivalent feature-scale information due to the combination effect.(2)Depthwise Convolution. Although there is no significant increase in the number of parameters required by the residual structure of Res2Net, its multi-branch structure destroys the parallelism of the network and reduces the speed of network computation. In order to avoid slowdown of the model, the bottleneck layer in the stage4 part of the convolution module of the ResNet network is replaced with the multi-branch convolution form from Res2Net. For the other stages, the optimization ideas verified in ConvNeXt [34] are borrowed for further improvement, and the depthwise convolution proposed in ResNeXt [35] is used to achieve a better balance between computational effort and accuracy. The structure of Res2Net combined with depthwise convolution is shown in Figure 9.(3)Inverse Bottleneck. In this paper, we use the inverse bottleneck layer structure proposed in MobileNetV2 [36], which adopts the form of “small dimension–large dimension–small-dimension” to ensure that the information of the feature map can avoid the information loss caused by compression of dimensionality when transforming between different dimensional feature spaces. This prevents the loss of detailed features on the body of the pig during the downsampling process.(4)Improved Normalization and Activation Layers. In this paper, we reduce the oscillation of gradients during training by reducing the number of normalization and activation layers, using layer normalization instead of batch normalization, and using GELU instead of ReLU.(5)Using 7 × 7 Convolution instead of 3 × 3 Convolution. In this study, we borrow the concept of the Swin Transformer [37] to increase the perceptual field and use 7 × 7 convolution kernels instead of 3 × 3 convolution kernels in ResNet, which results in a model with a larger perceptual field.

The structure of ResNet with the five improvements mentioned above is shown in Figure 10.

#### 2.4.2. The Pig Keypoint Detection Algorithm Based on Keypoint R-CNN Network

In addition to adhesion, multiple pigs passing through the aisle may block each other, and in certain images the body of the pig may be mutilated due to the location of the camera. In order to solve this problem, we define five key points on the pig: the head, neck, back, rump, and tail. We then combine these with knowledge on human pose recognition to analyze several common “mutilation” phenomena, as shown in Figure 11. After completion of image segmentation, the keypoint detection algorithm is used to detect the key points of the pig’s body in order to determine whether the pig’s body is complete or not, thereby eliminating images that are not suitable for body mass estimation.

The keypoint detection algorithm in this paper was built based on the Keypoint R-CNN network, which is an extension of the Mask R-CNN network that replaces the mask branch of the Mask R-CNN network with a keypoint detection branch. Keypoint detection includes detecting pigs and locating their keypoints; the network encodes the keypoints of the detected objects individually, and outputs the type and location of each keypoint. Similar to the instance segmentation algorithm, the keypoint monitoring algorithm needs to classify the pixels in the image while detecting individual pigs, which requires a backbone network with a strong ability to handle multi-scale information. Thus, in this paper the backbone network of the Keypoint R-CNN algorithm is replaced with the improved ResNet described in Section 3.1. Based on the principle of transfer learning, the trained Mask R-CNN algorithm model is used as the pre-trained model of Keypoint R-CNN algorithm to reduce the training time of the algorithm and the oscillation during the training process.

#### 2.4.3. Pig Mass Estimation Algorithm Based on Improved ResNet

The pig body mass estimation algorithm in this paper was based on the improved ResNet described in Section 3.1, with the fully connected layer of the network modified to regress a body mass value. The algorithm structure is shown in Figure 12. The pig body mass estimation algorithm takes processed pictures of pigs as the input and their actual body mass as the target value, which enables the network to learn the features needed for body mass estimation. In addition, it uses the weight model of the backbone network of the trained Keypoint R-CNN algorithm as the pre-training model of the body mass estimation algorithm.

## 3. Results

To verify the performance of the proposed PMEM-DLWC model, in this paper we implement the above algorithm based on Python, OpenCV, and other software libraries, and conduct model training and testing under the Ubuntu OS environment. The implementation was built on the basis of the following frameworks and packages: Python 3.9, Pytorch framework (version 1.9), MMPose framework (version 0.280), MMDetection framework (version 2.250), CUDA (version 10.0), cuDNN (version 8.0), and OpenCV (version 3.4). The hardware and software environments used for model training and testing were as follows: Ubuntu 18.04.6 LTS OS, Intel Core I7-9700 3.0 GHZ CPU, NVIDIA A30 GPU, 24G video memory, 16G RAM, 256G SSD + 8T mechanical hard disk.

### 3.1. Experiments and Results for Pig Instance Segmentation Algorithm

In this paper, a pig instance segmentation dataset was constructed for the training, validation, and testing of the instance segmentation algorithm. In total, 9800 images were included in the dataset, which was labeled using the Labelme data labeling tool and saved in COCO [38] format.

To evaluate the performance of the instance segmentation algorithm, this study uses Average Precision (AP), a common evaluation metric in the field of machine learning, which is mainly characterized by the Precision Rate (PR) and Recall Rate (RR). The Intersection Over Union (IOU) is used in the instance segmentation algorithm to distinguish whether the prediction results belong to positive or negative cases; the specific formulas for the four parameters IOU, AP, PR, and RR are shown below.
(5)$$\mathrm{IOU}=\frac{\mathrm{target}\cap \mathrm{prediction}}{\mathrm{target}\cup \mathrm{prediction}},$$
(6)$$\mathrm{AP}=\sum_{k=0}^{k=n-1}\left[\mathrm{Recalls}\left(k\right)-\mathrm{Recalls}\left(k+1\right)\right]\cdot \mathrm{Precision}\left(k\right),$$
(7)$$\mathrm{PR}=\frac{\mathrm{TP}}{\mathrm{TP}+\mathrm{FP}},$$
(8)$$\mathrm{RR}=\frac{\mathrm{TP}}{\mathrm{TP}+\mathrm{FN}},$$
where target denotes the labeling mask, prediction denotes the prediction mask, TP denotes the sample size of positive cases correctly classified, TN denotes the sample size of negative cases correctly classified, FP denotes the sample size of negative cases incorrectly classified as positive, FN denotes the sample size of positive cases incorrectly classified as negative, *n* denotes the number of IOU thresholds, and Recalls(*k*) and Precision(*k*) denote the recall rate and precision rate when the *k_th_* IOU threshold is taken, respectively.

In this paper, the pre-trained model trained on the COCO dataset is used to initialize the instance segmentation algorithm model, and the algorithm training parameters are set as shown in Table 1.

To verify the improvement effect of the algorithm, this experiment uses ResNet, ConvNeXt, and improved ResNet as the backbone network of the instance segmentation algorithm for training, validation, and testing. The loss function curves on the training and validation sets during algorithm training are shown in Figure 13, and the specific results are shown in Table 2. The solid lines in the figure represent the loss function convergence curves of the instance segmentation algorithms with different backbone networks when trained on the training set. The dashed line indicates the loss function convergence curve of the algorithm on the validation set at the same moment. The improved ResNet network converges faster than the other network and the difference between the loss values in the training set and the loss values in the validation set after convergence is smaller, which indicates that the model has a better fit. The parameters in the improved ResNet network are 3.16M larger than in the Resnet network and 0.13M larger than in ConvNeXt. The average segmentation speed of the image is 0.15 s frame^−1^, which is 0.065 s frame^−1^ and 0.071 s frame^−1^ lower than that of the instance segmentation algorithm with ConvNeXt and ResNet as the backbone network, respectively. The average accuracy on the test set is 91.04%, which is 2.06% and 3.19% higher than the instance segmentation algorithm with ConvNeXt and ResNet as the backbone network.

### 3.2. Experiments and Results for Pig Keypoint Detection Algorithm

In this paper, keypoints are labeled based on an instance segmentation dataset to construct a keypoint detection dataset for the training, testing, and validation of the pig keypoint detection algorithm.

In terms of algorithm evaluation methods, the keypoint detection algorithm uses the same performance evaluation criterion as the instance segmentation algorithm, that is, the AP. Unlike the instance segmentation algorithm, which uses IOU evaluation, the keypoint detection algorithm uses Object Keypoint Similarity (OKS) to define the matching degree between the real object and the predicted object. Similar to IOU, the prediction is considered correct only when the matching degree between the predicted object and the real object exceeds a certain threshold, and the OKS is calculated by the formula
(9)$$\mathrm{OKS}=\exp(-\frac{d_{i}^{2}}{2s^{2}k_{i}^{2}}),$$
where *d_i_* denotes the Euclidean distance between the predicted keypoint and the real keypoint for a given pixel, *s* denotes the square root of the area size occupied by the target where the keypoint is located on the image in pixel^2^, and *k* denotes the normalization factor of the *k_th_* keypoint.

The training parameters of the keypoint detection algorithm are set the same as in the instance segmentation algorithm. The loss function curves of the algorithm on the training and validation sets during training are shown in Figure 14, and the detailed results are shown in Table 3. The solid lines in the figure represent the loss function convergence curves of the keypoint detection algorithms with different backbone networks when trained on the training set. The dashed line indicates the loss function convergence curve of the algorithm on the validation set at the same moment. The results show that the algorithm performs best when the improved ResNet network is used as the backbone network. The keypoint detection accuracy on the test set is 89.59%, which is 1.05 and 4.08 percentage points higher than the keypoint detection algorithm with ConvNeXt and Res2Net as the backbone network.

### 3.3. Experiments and Results for Pig Body Mass Estimation Algorithm

In this paper, a pig body mass dataset containing body mass data and images of 117 pigs with a total of 6384 images was constructed for training, validation, and testing of the pig body mass estimation algorithm.

To evaluate the performance of the pig body mass estimation algorithm, the Root Mean Square Error (RMSE) and Mean Absolute Percentage Error (MAPE) were used as evaluation metrics. RMSE is the square root of the ratio of the square of the deviation of the predicted value from the true value to the number of observations m; it measures the deviation between the predicted value and the true value, and is more sensitive to outliers in the data. MAPE represents the mean of the absolute percentage error of each data instance in a dataset; it indicates the accuracy of the predicted value compared to the true value. The specific calculation formula is as follows:(10)$$\mathrm{RMSE}(W^{p},W^{r})=\sqrt{\frac{1}{m}\sum_{i=1}^{m}{\left(W_{i}^{p}-W_{i}^{r}\right)}^{2}},$$
(11)$$\mathrm{MAPE}=\frac{100\%}{m}\times \sum_{i=1}^{m}\left|\frac{W_{i}^{p}-W_{i}^{r}}{W_{i}^{r}}\right|.$$

Here, *W_p_* denotes the pig body mass predicted by the model, *W_r_* denotes the real pig body mass in kg, and *m* denotes the numerical value of pig body mass in the dataset.

The algorithm training parameters are set as shown in Table 4.

The loss function curves on the training and validation sets during the training of the body mass determination algorithm for different backbone networks are shown in Figure 15. The solid lines in the figure represent the loss function convergence curves of the mass estimation algorithms with different backbone networks when trained on the training set. The dashed line indicates the loss function convergence curve of the algorithm on the validation set at the same moment. The RMSE values after convergence of the improved ResNet network are significantly lower than those of the other two networks; the model with the best performance on the validation set was selected as the final algorithm model for this algorithm. The RMSE, MAPE, and average predicted speed per image of the algorithm on the test set are shown in Table 5. The newly designed network in this paper has a slightly decreased computing speed than the other two networks, and the RMSE on the test set is 3.52 kg, which is 7.35 kg and 1.63 kg lower than the pig body mass determination algorithm with ResNet and ConvNeXt as the backbone networks, respectively.

## 4. Discussion

Figure 16 shows the distribution of estimated body mass and actual body mass obtained with the pig body mass estimation model for all the images of pigs in the test set; it can be seen that the data are evenly distributed in each mass interval and the estimation errors are similar. Despite the relatively poor quality of the data obtained in the aisle environment, the model achieved good results on the test set, with an RMSE of 3.52 kg and MAPE of 2.82%. The model is built based on a convolutional neural network that estimates the body mass of pigs in an “end-to-end” way, using pre-processed images as input and producing a final output corresponding to the body mass, thereby eliminating the process of feature engineering in traditional machine learning methods. The average estimation speed of the model is 0.339 s per image, which is suitable for aisle scenarios in which there are strict limits on the speed of acquiring the body masses of pigs. Figure 17 shows an example of the application of the body mass estimation model in the practical scenario.

In the estimation results of the test set, there are six images with an estimation error greater than 10 kg, as shown in Figure 18. This problem can be solved by designing a target tracking algorithm to track the pigs and capture the images only when the pigs appear in the central area, or alternatively by taking the positions of the pigs in the images as parameters together with the images and inputting them into the body mass estimation algorithm to obtain more accurate measurements of the body mass of the pigs. We intend to improve the algorithm in subsequent work to further improve the proposed body mass estimation model.

## 5. Conclusions

To address problems with unstable image quality, inconspicuous boundaries between pigs, and frequent mutual occlusion of pigs in aisle environments, which can interfere with the estimation of pig body mass, this paper introduces a series of data processing algorithms based on computer vision technology, including blurry detection, a similarity metric, instance segmentation, and keypoint detection. Using these, we build the proposed PMEM-DLWC model to solve the problem of pig body mass determination in aisle environments. The main research work and innovations in this paper include the following aspects.

(1)In this paper, an unconstrained pig body mass estimation model is designed. The model uses blurry detection and similarity metric algorithms to ensure that the image quality used for body mass estimation is stable, then an instance segmentation algorithm is used to extract individual pigs in the image. A keypoint detection algorithm is used to eliminate the obscured pigs, and finally the image is fed into the body mass estimation algorithm to obtain the body mass.(2)In this paper, the proposed PMEM-DLWC model is improved by combining the latest developments in deep learning in recent years. The ResNet backbone network was improved by introducing multi-branch convolution, depthwise convolution, and an inverse bottleneck to construct a network that achieves a good balance between the number of model parameters and the accuracy rate. The RMSE of the model on the test set was 3.52 kg, and the average estimation speed was 0.339 s per image. These result show that the proposed mass estimation model can automatically extract the important features required for mass estimation. The reported speed and accuracy are effectively able to meet actual production needs.(3)The proposed PMEM-DLWC model can be used to estimate the body mass of pigs in an unconstrained environment. Compared with other methods, the pig body mass estimation model constructed in this paper has fewer constraints on pig posture as well as on the data acquisition environment, and the accuracy of the body mass estimation model is further improved by modifying the backbone network while maintaining low-cost equipment. The proposed approach can provide data and technical support for the determination of pig body mass, pig grading management, analysis of pig weight gain patterns, and realization of pig grading management, and can provide new ideas and methods for future research about precision livestock farming.

In future applications, mass determination could be enhanced by utilizing multiple images or videos of pigs in different postures. This approach would better capture the complete information of obscured pigs, including body height information, which can be challenging to obtain from an overhead view. By using multiple images instead of a single image for weight determination, the accuracy of the mass estimation model could be improved while maintaining a low-cost device. Moreover, the algorithm has the potential to be extended to a wider range of applications in combination with an object tracking algorithm. This would enable the algorithm to be used in a group breeding environment, facilitating the achievement of a wider range of application goals. Finally, because the PMEM-DLWC model constructed in this paper contains multiple tasks, future work will need to consider the issue of how to perform multi-objective optimization of the model during training.

## Figures and Tables

**Figure 1 animals-13-01376-f001:**
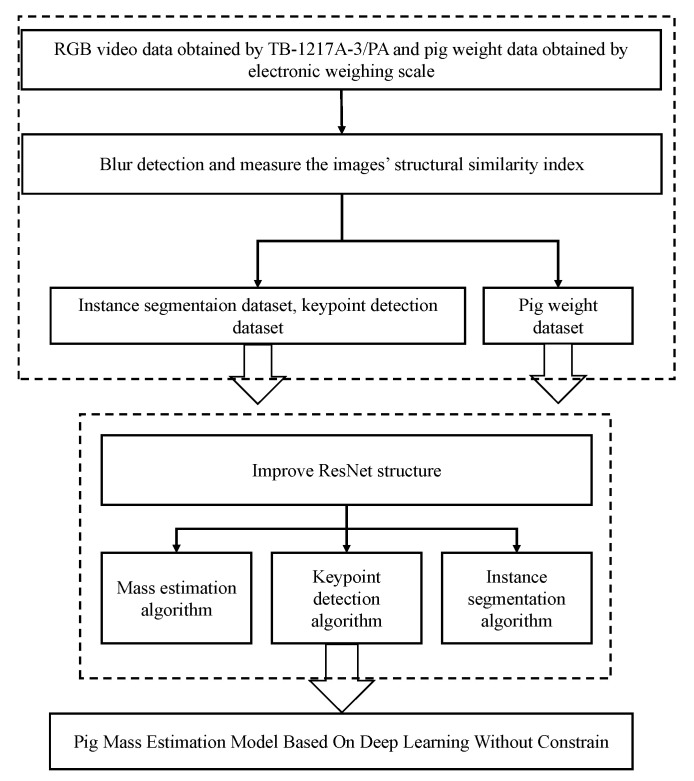
The technical roadmap of this paper.

**Figure 2 animals-13-01376-f002:**
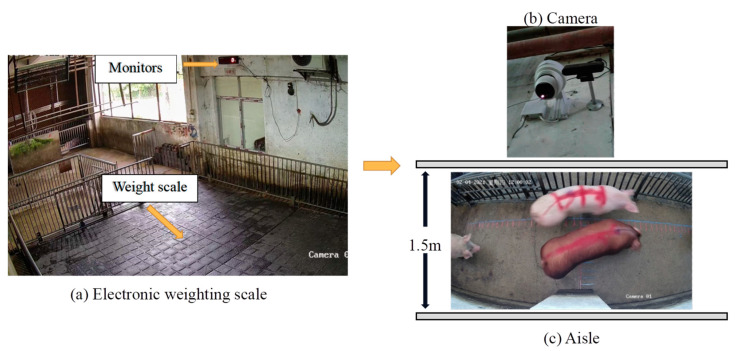
Experimental setup: The pigs’ mass was weighted at the electronic weighing scale, and after weighing the pigs were driven to the aisle for video collection. The camera was installed above the aisle, with the height between the ground and the camera being 2.8 m, and the width of the aisle 1.5 m. The camera was connected to a micro-computer for data transfer. (**a**) Electronic weighing scale with 0.1 kg accuracy used for weighing the pigs. (**b**) Hikvision TB-1217A-3/PA binocular camera. (**c**) Aisle used as pig entry channel in the slaughterhouse, with a width of 1.5 m.

**Figure 3 animals-13-01376-f003:**
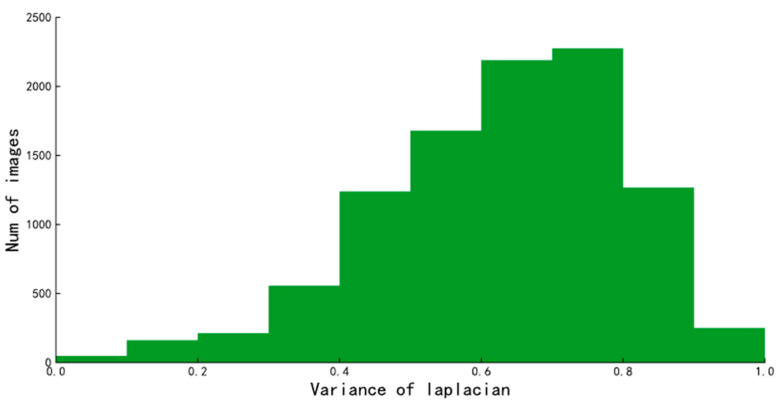
Distribution of the Laplacian function values of the video frames.

**Figure 4 animals-13-01376-f004:**
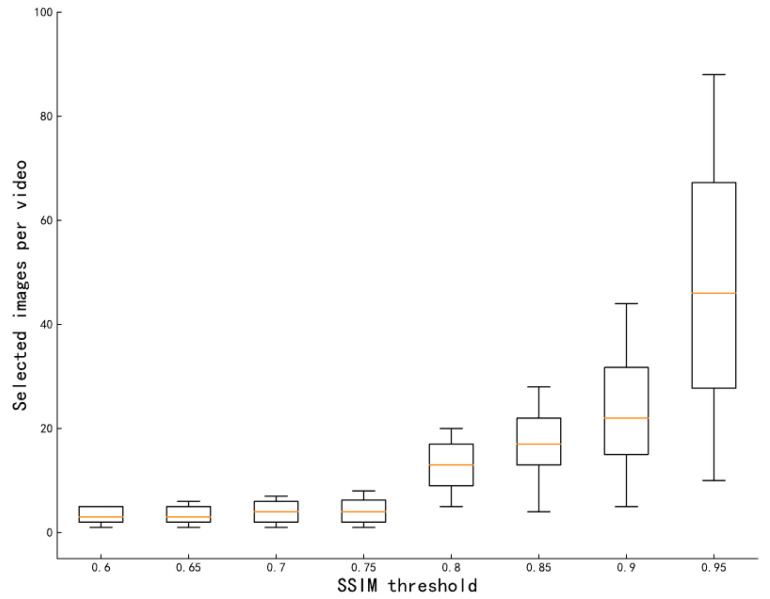
Influence of different SSIM threshold values on the number of selected images per video. For each video, the SSIM value between the selected video frames must not exceed the SSIM threshold. The figure shows the distribution of selected images per video depending on the SSIM threshold.

**Figure 5 animals-13-01376-f005:**
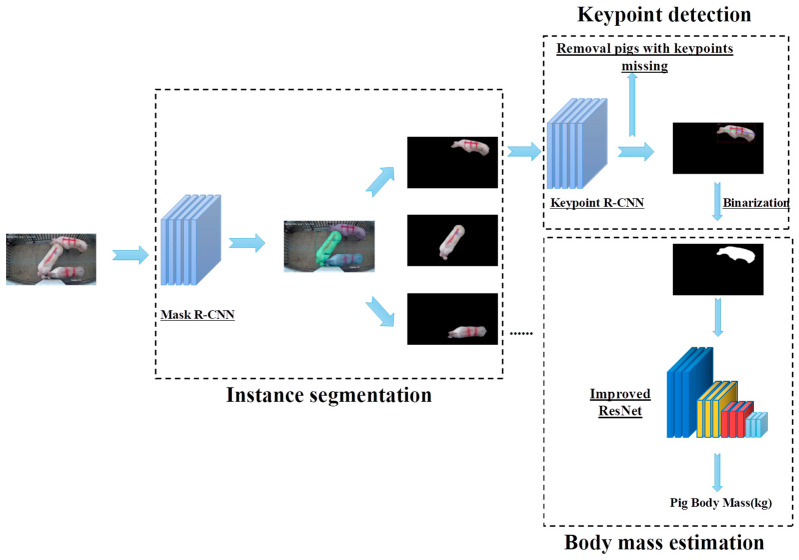
Structure of Pig Mass Estimation Model based on Deep Learning Without Constraint. Pigs are segmented from images by the instance segmentation algorithm, and the keypoints are checked by the keypoint detection algorithm. Finally, the mass estimation algorithm outputs the estimated weight for a pig based on the related binarized image.

**Figure 6 animals-13-01376-f006:**
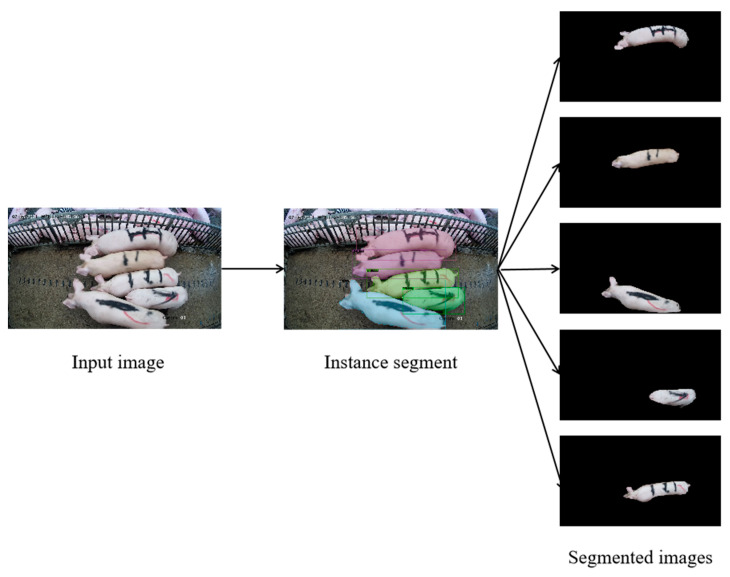
Example of the instance segmentation algorithm. Pigs in the aisle are segmented from the background using the instance segmentation algorithm.

**Figure 7 animals-13-01376-f007:**
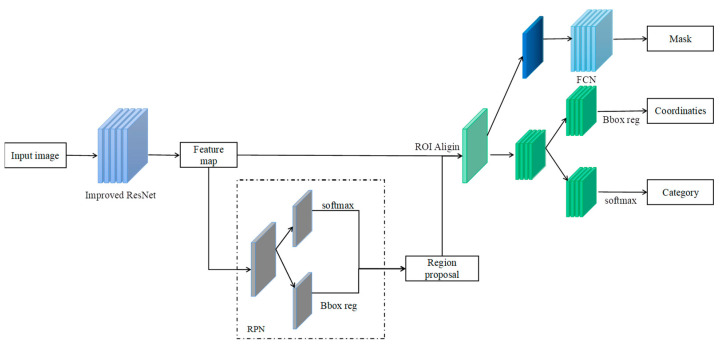
Structure of Mask R-CNN algorithm.

**Figure 8 animals-13-01376-f008:**
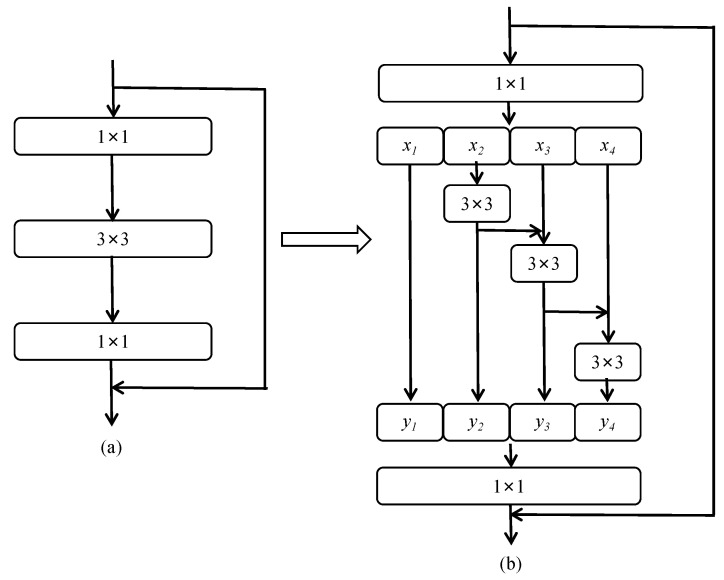
(**a**) Residual structure of ResNet and (**b**) Residual structure of Res2Net. Here, *x_1_*~*x_4_* denote the input feature maps after grouping; each feature subset *x_i_* has the same spatial size and 1/4 the number of channels compared with the input feature map. Except for *x_1_*, each *x_i_* has a corresponding 3 × 3 convolution; *y_1_*~*y_4_* denote the feature map after feature extraction by 3 × 3 convolution.

**Figure 9 animals-13-01376-f009:**
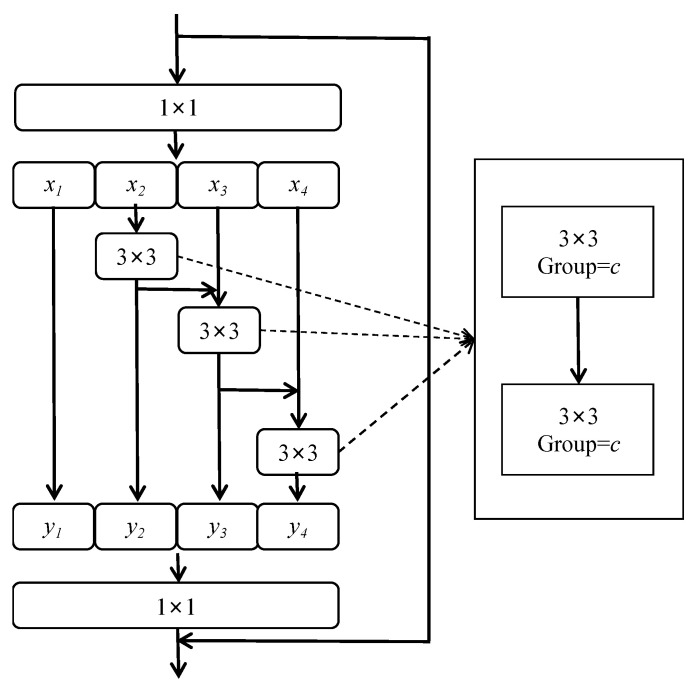
Residual structure of Res2Net combined with depthwise convolution; 3 × 3 convolution is displaced by group convolution, and the number of groups is equal to the number of channels in the input feature map.

**Figure 10 animals-13-01376-f010:**
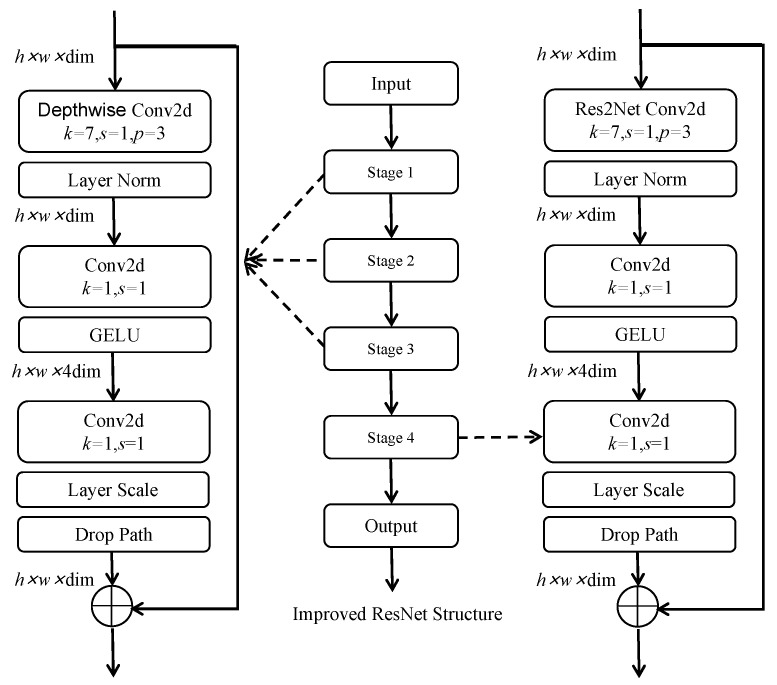
Structure of our improved ResNet network; h and w are the length and width of the input feature maps, dim is the number of channels with the input feature maps, k means the convolution kernel size, s means the stride of the convolution kernel, and p means the padding of the convolution kernel. To reduce the number of parameters, the improved ResNet network does not introduce the multi-branch convolution module in the first three stages, only in the fourth stage.

**Figure 11 animals-13-01376-f011:**
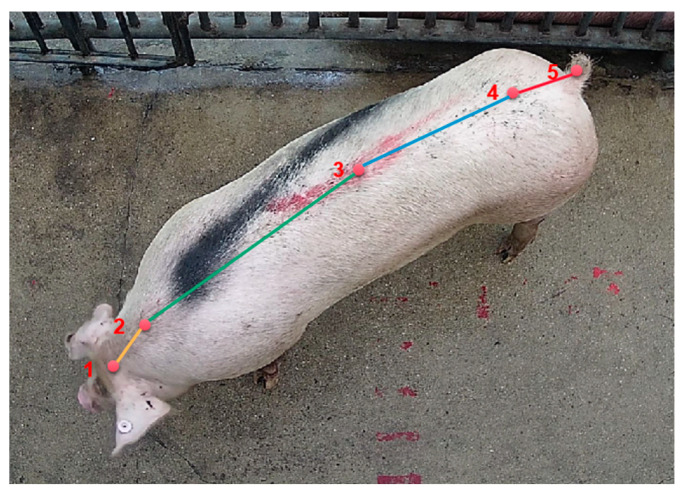
Design of keypoints in our pig keypoint dataset, including: 1, head; 2, neck; 3, back; 4, hip; and 5, tail.

**Figure 12 animals-13-01376-f012:**
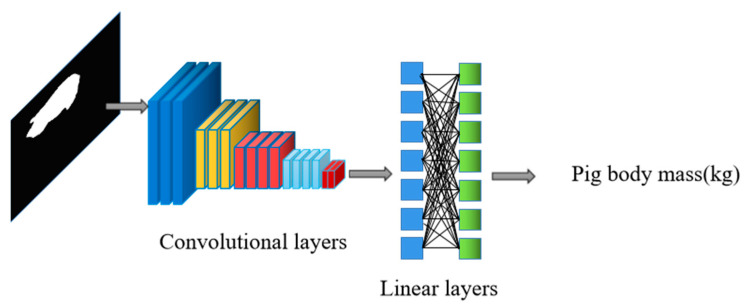
Structure of pig mass estimation algorithm.

**Figure 13 animals-13-01376-f013:**
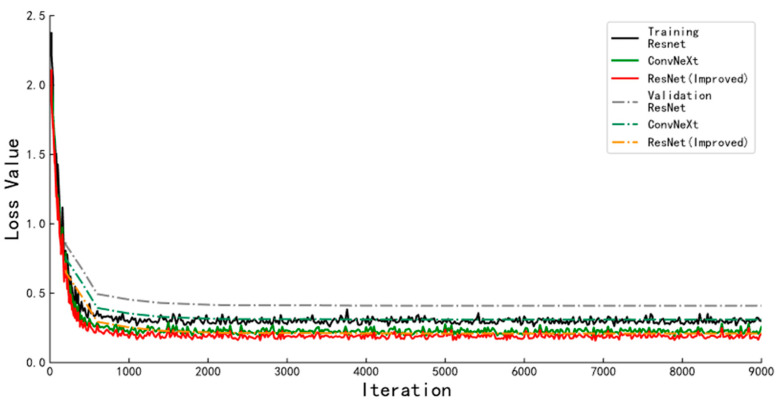
Loss function curves of instance segmentation algorithms with different backbone networks.

**Figure 14 animals-13-01376-f014:**
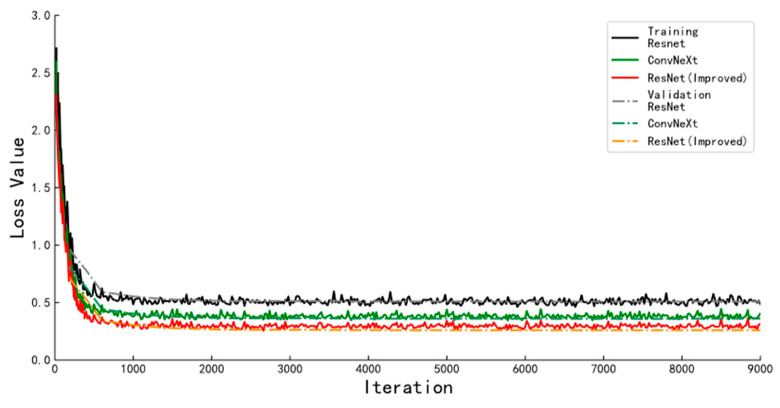
Loss function curves of keypoint detection algorithms with different backbone networks.

**Figure 15 animals-13-01376-f015:**
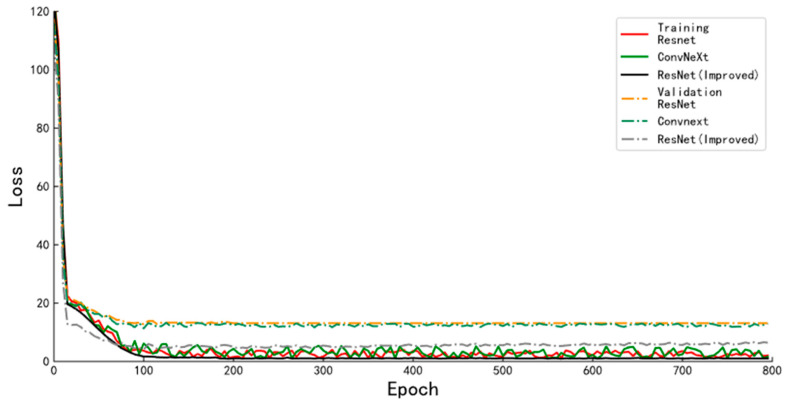
Loss function curves of the mass estimation algorithm for different backbone networks.

**Figure 16 animals-13-01376-f016:**
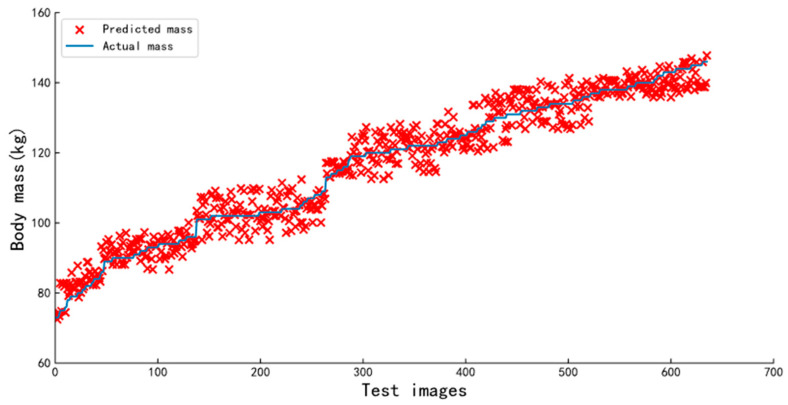
The actual mass and corresponding predicted mass for the 636 test images.

**Figure 17 animals-13-01376-f017:**
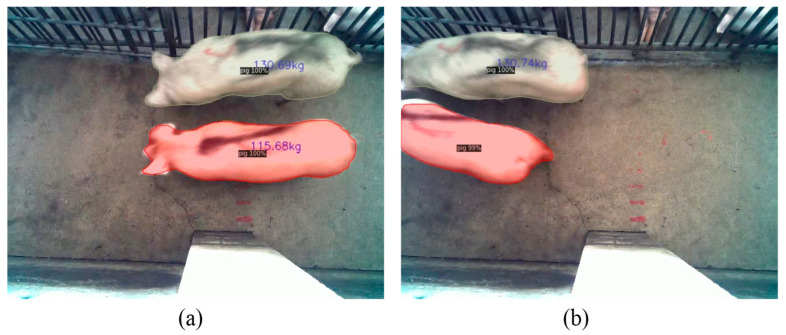
An example of the application of the pig body mass estimation model in a practical scenario where two pigs are passing through the aisle side by side. The bodies of both pigs are clearly visible in Figure (**a**), and the model estimates the body masses of both pigs separately. In Figure (**b**), the head of one pig is out of camera range, and the model does not estimate its weight.

**Figure 18 animals-13-01376-f018:**
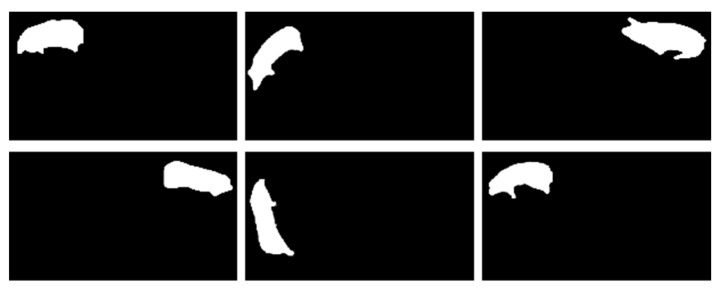
The worst mass estimation result on the test dataset; note that the pigs in these images are all at the boundary of the image.

**Table 1 animals-13-01376-t001:** Parameters of pig instance segmentation algorithm.

Parameters	Value
learning rate	0.0001
gamma	0.1
weight decay	0.05
batch size	4
iteration	9000

**Table 2 animals-13-01376-t002:** Detailed results of instance segmentation algorithm with different backbone networks, including the average precision of the bounding box and mask, average speed, and model size.

BackBone Network	Training Set		Validation Set		Testing Set		Average Speed (s Frame^−1^)	Model Size
	Average Precision of Bounding Box	Average Precision of Mask	Average Precision of Bounding Box	Average Precision of Mask	Average Precision of Bounding Box	Average Precision of Mask		
ResNet	97.22%	90.59%	93.93%	88.39%	92.84%	87.85%	0.079	25.56 M
ConvNeXt	98.85%	91.60%	94.10%	89.88%	93.03%	88.98%	0.085	28.59 M
Improved ResNet	99.81%	92.30%	96.63%	91.65%	93.65%	91.04%	0.15	28.72 M

**Table 3 animals-13-01376-t003:** Detailed results for the keypoint detection algorithm with different backbone networks, including the average precision of the bounding box and keypoint and average speed.

BackBone Network	Training Set		Validation Set		Testing Set		Average Speed (s Frame^−1^)
	Average Precision of Bounding	Average Precision of Keypoint	Average Precision of Bounding	Average Precision of Keypoint	Average Precision of Bounding	Average Precision of Keypoint	
ResNet	94.45%	87.94%	93.50%	85.59%	89.75%	85.51%	0.065
ConvNeXt	97.86%	89.59%	94.52%	89.62%	92.53%	88.54%	0.087
Improved ResNet	98.81%	91.89%	96.03%	90.89%	94.38%	89.59%	0.12

**Table 4 animals-13-01376-t004:** Parameters of pig mass measurement algorithm.

Parameters	Values
learning rate	0.0025
gamma	0.1
weight decay	0.0005
batch size	1
epoch	800

**Table 5 animals-13-01376-t005:** Detailed results for the mass estimation algorithm with different backbone networks, including the root mean square error, mean absolute percentage error, and average speed.

BackBone Network	Training Set		Validation Set		Testing Set		Average Speed (s Frame^−1^)
	Root Mean Square Error	Mean Absolute Percentage Error	Root Mean Square Error	Mean Absolute Percentage Error	Root Mean Square Error	Mean Absolute Percentage Error	
ResNet	5.94 kg	3.38%	9.36 kg	6.19%	10.87 kg	6.77%	0.014
ConvNeXt	3.65 kg	3.15%	4.94 kg	3.93%	5.15 kg	4.12%	0.052
Improved ResNet	2.81 kg	2.25%	3.01 kg	2.41%	3.52 kg	2.82%	0.069

## Data Availability

Due to an agreement with our collaborators, the dataset cannot be made public now.

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
