# Peer review of "A Pig Mass Estimation Model Based on Deep Learning without Constraint"

_animals, 2023, doi:10.3390/ani13081376_

Round 1
Reviewer 1 Report
-Line 210 - Figure 1 should be Figure 3
-The paper should be interesting ;;;
-it is a good idea to add a block diagram of the proposed research/review (step by step);;;
-What is the result of the analysis?;;
-figures should have high quality. ;;;;;
-text should be formatted;;;;
-references should be formatted;;;
-please add photos of the application of the proposed research, 2-3 photos ;;;
-what will society have from the paper?;;
-labels of figures should be bigger;;;; - Figure 1-3;;;
-Is there a possibility to use the proposed research, methods for other topics, deep learning etc.;;;
-please compare the advantages/disadvantages of other approaches of fault diagnosis;;;
-references should be from the web of science 2020-2023 (50% of all references, 30 references at least);;;
-Conclusion: point out what have you done;;;;
-please add some sentences about future work;;;
Author Response
Dear Editor and Reviewers,
Thanks very much for the time to review this manuscript. I appreciate all your comments and suggestions. These comments have been constructive in revising and improving our paper and have been an essential guide for our research. We have carefully studied the comments and made corrections that we hope will be approved. The revised sections are marked in the paper. The major corrections in the paper and responses to the reviewers' comments are in the attachment.

Reviewer 2 Report
I had the pleasure of reviewing the manuscript titled: “A Pig Mass Estimate Model based on Deep Learning Without Constrain.” The authors present a new model to accurately estimate pig mass from images and videos of pigs engaging in unconstrained activity in captive setting. They combine three deep neural networks that perform separate tasks in the process of estimating masses of pigs. The first network performs segmentation of different instances of pig, i.e., detecting and identifying different individuals in a frame and separating them for the other networks. The second deep network performs pose estimation in order to determine whether the segmented instance from the previous network contains the complete individual so as to remove the images that don’t contain the full body of an individual. The final network performs the mass estimation with decent accuracy.
The manuscript is generally well-written, with proper grammar and hardly any typos or grammatical errors. The authors avoid being unnecessarily verbose, which is good. The methods and analyses are of sufficient rigor for the task at hand. Their original contribution lies in the modifications they make to the network architectures to perform the unique task of accurately estimating mass of pigs. They also perform rigorous data cleaning steps to ensure retaining only the high-quality images from videos that contain frames of varied quality, most of which being unsuitable for mass estimation.
Given these qualities, I only have some minor suggestions for revising and one major suggestion.
1) The major issue is that neither the code, nor the model, nor the dataset is made available to the public. It is difficult to evaluate the merits of a paper that presents a new network and dataset without accessing the code and dataset themselves. If they want to keep their data and model private, at minimum they should include a demo of the model at work on a new video, i.e., on a video not included for training.
2) Most of the figures include text in small font sizes that is difficult to read. Please reformat the axes labels, ticks, captions, and legends such that they are readable when printed on a US letter or A4 paper sizes. The figures are also low quality. Perhaps using a vectorized format such as SVG or PDF would result in better quality figures.
3) The discussion section is inadequate and feels rushed. There isn’t sufficient discussion on how this work improves over other similar studies. Furthermore, there is no discussion about how other researchers could use this in their own work. Right now, it seems like a network with a unique application only in their captive setting. It would be useful to have more discussion on what other contexts and settings could utilize this model. For example, would it work for other species? What other applications could this model be extended to? For instance, would it work, with slight modifications in one of the networks, for other applications such as behavior recognition that could be relevant for monitoring health? The lack of code and dataset makes it impossible to evaluate whether other researchers could use it in their work. Other relevant details to include would be the annotation and output formats, though that information could also be on a GitHub page.
Overall, the model and the dataset could potentially be very useful for other researchers, so I would be happy to accept the paper once these changes are made and especially if the code and dataset are available on GitHub.
Author Response
Dear Editor and Reviewers,
Thanks very much for the time to review this manuscript. I appreciate all your comments and suggestions. These comments have been constructive in revising and improving our paper and have been an essential guide for our research. We have carefully studied the comments and made corrections that we hope will be approved. The revised sections are marked in the paper. Here is a demo video https://github.com/morningbearscau/pig/blob/main/out-1.mp4

Reviewer 3 Report
Following are my comments and suggestions:
1. Abstract: It should be written a very precise way.
2. Materials and Methods: Mathematical formulation of the model can improve the quality of the article.
3. Differences between original ResNet network and improved ResNet network are not explained clearly.
4. Improve the visual quality of Figs. 11-14.
5. Experimental results are satisfactory.
6. Authors can correlate their study with recently published article in image classification and segmentation as:
https://doi.org/10.1016/j.advengsoft.2022.103370
https://doi.org/10.1016/j.knosys.2021.107432
Author Response

(The authors gave the same response as above.)

Reviewer 4 Report
General observation
Abstract are too long. Abstract must be a single paragraph of about 200 words maximum.
Keywords: use ";" not ","
Missing space between text and [ for example line 40 "status[1,2]," also in English we don't use comma before and.
Correct missing space in whole document.
You have too long paragraphs correct this.
Use template indication in paper.
Correct text from "Table" and from "Figure" to respect template indication. For example Alignment must be Justified; Indentation left 46 mm, Right 0 mm; Spacing before 6pt and after 12 pt and font size 9
When you use equations. figures and table you must give and some explication text you use "as show". For example figure 13, what represent the yellow line? In table are indicate like be "Ournet"...
In English, you must put a comma before “and” when it connects two independent clauses.
Line 110 missing "." at the end of line.
Figure 1 You have 3 images please give name a,b and c and some explication.
Line 184 Missing text before equation
Line 190 try to move to next page.
Line 195 if you say "as follows" why you use "." and not ":"
Line 197 Missing some text before "x and y"
Line 210 correct to "Figure 3"
Line 273 Missing space "1)Multi-branch"
From line 245 correct text in "Justify" style
Line 312 is the last line in page 9 try to move in the next page. Also missing some explication text for figures 7 and 8
Line 357 "Figure 10. Structure of pig mass estimation algorithm." must be in the same page with figure
389 Remove text "This is a table. Tables should be placed in the main text near to the first time they are cited." and add the correct text.
390 missing space between table and text
393 add some descripted text for figure aa and for table 2 /// same for line 426 Figure 12 and table 3
409 "3.2. Experiment and result of pig keypoint detection algorithm" at my pdf it's last line on page move this line to next page.
453 correct Fig. to Figure
Author Response

(The authors gave the same response as above.)

Round 2
Reviewer 1 Report
please add arrows to figures what is what
Author Response
Dear Editor and Reviewers,
Thanks very much for the time to review this manuscript. I appreciate all your comments and suggestions. These comments have been constructive in revising and improving our paper and have been an essential guide for our research. We have carefully studied the comments and made corrections that we hope will be approved. The revised sections are marked in the paper. The major corrections in the paper and responses to the reviewers' comments are as follows.

Reviewer 4 Report
Abstract is more than 200 words
Line 320 move in he same page with figure
Missing space before line 403
Author Response
Dear Editor and Reviewers,
Thank you so much for taking the time to review this manuscript. I really appreciate all your comments and suggestions. These comments have been very helpful in revising and improving our thesis and have been an important guide to our research. Please forgive our lack of detail in the original manuscript regarding layout and formatting, etc. In the revised manuscript, we have improved the formatting of the entire paper. We have carefully studied the comments and amended the formatting section of them, and hope that we have fully appreciated the experts' comments and responded accurately. The revised sections are marked in the thesis. The main changes in the thesis and the responses to the reviewers' comments are listed below.
